# Multidrug-Resistant Bacteria Contaminating Plumbing Components and Sanitary Installations of Hospital Restrooms

**DOI:** 10.3390/microorganisms12010136

**Published:** 2024-01-10

**Authors:** Felice Valzano, Anna Rita Daniela Coda, Arcangelo Liso, Fabio Arena

**Affiliations:** 1Department of Clinical and Experimental Medicine, University of Foggia, Via Napoli 20, 71122 Foggia, Italy; felice.valzano@unifg.it (F.V.); fabio.arena@unifg.it (F.A.); 2Department of Medical and Surgical Sciences, University of Foggia, Via Napoli 20, 71122 Foggia, Italy; daniela.coda@unifg.it; 3IRCCS Don Carlo Gnocchi Foundation, Via di Scandicci 269, 50143 Florence, Italy

**Keywords:** multidrug resistance, toilet, plumbing systems, bioaerosol, droplets, hospital, contamination, infections

## Abstract

Antimicrobial resistance (AMR) poses several issues concerning the management of hospital-acquired infections, leading to increasing morbidity and mortality rates and higher costs of care. Multidrug-resistant (MDR) bacteria can spread in the healthcare setting by different ways. The most important are direct contact transmission occurring when an individual comes into physical contact with an infected or colonized patient (which can involve healthcare workers, patients, or visitors) and indirect contact transmission occurring when a person touches contaminated objects or surfaces in the hospital environment. Furthermore, in recent years, toilets in hospital settings have been increasingly recognised as a hidden source of MDR bacteria. Different sites in restrooms, from toilets and hoppers to drains and siphons, can become contaminated with MDR bacteria that can persist there for long time periods. Therefore, shared toilets may play an important role in the transmission of nosocomial infections since they could represent a reservoir for MDR bacteria. Such pathogens can be further disseminated by bioaerosol and/or droplets potentially produced during toilet use or flushing and be transmitted by inhalation and contact with contaminated fomites. In this review, we summarize available evidence regarding the molecular features of MDR bacteria contaminating toilets of healthcare environments, with a particular focus on plumbing components and sanitary installation. The presence of bacteria with specific molecular traits in different toilet sites should be considered when adopting effective managing and containing interventions against nosocomial infections potentially due to environmental contamination. Finally, here we provide an overview of traditional and new approaches to reduce the spreading of such infections.

## 1. Introduction

In recent years, the WHO has identified antimicrobial resistance (AMR) as one of the top 10 global public health threats facing humanity [1]. In high-income countries, infections by multidrug-resistant (MDR) bacteria are mainly contracted in the healthcare setting and are responsible for a high proportion of deaths and disability-adjusted life years [2,3]. On the other hand, *Clostridioides difficile*, which is a non-MDR pathogen, is another urgent threat to public health and the leading cause of healthcare facility-associated infections, which occur due to increased and inappropriate use of broad-spectrum antibiotics [4,5].

In the healthcare setting, infected and/or colonized patients are the most important MDR bacteria reservoir, with healthcare workers and visitors playing a role in transmission [6,7]. Furthermore, MDR bacteria can be found in the healthcare environment, including dry surfaces in patient care areas, medical devices, dust and wet surfaces, and moist environments [6,7].

Recent studies indicate that sinks and other drains, such as toilets or hoppers, in healthcare facilities can become contaminated with MDR bacteria [8]. MDR pathogens can grow as biofilms into pipes and persist in drains for long time periods and are virtually impossible to fully remove. Moreover, because different bacterial species may contaminate the same drain, drains could serve as sites where antibiotic resistant genes are transferred between species [8,9,10,11].

The transmission of MDR bacteria from sinks and other drains, toilets or hoppers to patients can occur through contact (i.e., touching), sprays and splashes, and inhalation [8] (Figure 1).

A contaminated toilet hence represents a milieu where infectious bioaerosols and droplets, generated during and between uses, potentially expose toilet users, cleaners, plumbers and healthcare workers to pathogens [12,13,14,15].

Bioaerosol production within toilets was first reported in literature in 1955 by Jessen [16], and subsequent studies have deepened knowledge of this generally neglected problem. The production of particles following the use of the toilet flush has been measured through air sampling by Knowlton et al. [17]. The authors demonstrated that the production of rebound droplets increases without the closing of the toilet lid during flushing [17].

Fluid dynamics simulations demonstrated that virus aerosol particles may be massively upward transported due to toilet flushing, with 40–60% of particles rising above the toilet seat and reaching to a height of 106.5 cm from the ground [18].

The aim of this review is (i) to summarize studies reporting the isolation of MDR bacteria from hospital restrooms (with a particular focus on plumbing components and sanitary installations situated in toilets) and to describe their molecular traits; (ii) and to describe effective interventions to prevent MDR dissemination from the restrooms or to terminate outbreaks. Traditional and innovative approaches to reduce contamination risks are also briefly discussed.

## 2. Research Methodology

This literature review was performed without time limits (last accessed: 10 November 2023). A comprehensive search strategy was deployed, encompassing multiple academic databases, including PubMed, Scopus and Web of Science. Updates from the World Health Organization (WHO) and other governmental organizations around the world (Italian National Institute of Health (ISS), European Center for Disease Prevention and Control (ECDC), American Center for Disease Control and Prevention (CDC), United States Department of Labor) were also examined. Google Patents was queried to search for patents and patent applications that were useful and pertinent to the topic of this review (last accessed 10 November 2023).

The search terms used were “bacteria”, “multidrug resistance”, “toilet”, “bioaerosol”, “droplets”, “hospital”, “contamination” and “infections”. Titles and abstracts were screened for duplicates, and duplicates were excluded. Following this selection, full texts were analysed. Articles not compliant with the aim of this review were excluded mainly for the following reasons: (i) studies concerning settings other than the hospital; (ii) studies not related to multidrug-resistant bacteria; (iii) unfinished/unpublished studies. Conference abstracts were also excluded.

## 3. Contamination of Different Sites in Hospital Toilets by MDR Bacteria

### 3.1. Contamination of Toilet Bowl, Seat and Rim

Several studies have shown a correlation between hospital-acquired infections caused by MDR and contamination of different parts of the toilet (Table 1). In most cases, such a relation was the cause of outbreak persistence or recurrence and represented a serious concern for patient clinical outcomes.

A study by Smismans and colleagues reported on the first detection of carbapenemase-producing *C. freundii* in toilet bowls and the subsequent hospital-associated transmission to patients in one hospital ward [19]. By sampling of high-touch surfaces, medical tools and toilet water, the authors found the latter to be positive for *C. freundii* carrying the *bla*_OXA-48_ gene (Table 1). Admission in the room where OXA-48-producing *C. freundii* was found in the toilet water was a risk factor for infection. Toilet cleaning with biguanide/quaternary ammonium for 15 min, followed by disinfection with 2500 parts per million (ppm) peracetic acid for 30 min was effective in eradicating the strain from the toilet and terminating the outbreak [19].

Similarly, environmental investigations at a haematological ward of a French university hospital identified seven of 74 toilet rims positive for *bla*_OXA-48_-positive CPE, including *C. freundii*, *Enterobacter sakazakii* and *E. coli* [20]. Transmission of clonally related OXA-48-producing *C. freundii* from the hospital environment to patients was demonstrated by whole genome comparisons. The majority of the clinical and environmental isolates of OXA-48-producing *C. freundii* belonged to ST22 and were considered highly related by genome sequencing analysis (<50 SNPs between different strains) [20] (Table 1). It is interesting to note that, despite the concomitant circulation of NDM-type-producing Enterobacterales in the same ward, only *bla*_OXA-48_-positive CPE were found in toilets, while strains harbouring *bla*_NDM-type_ genes were obtained from human samples only, demonstrating a higher environmental persistence ability of *bla*_OXA-48_-positive organisms as compared to other CPE.

Among MDR bacteria colonizing the gut of inpatients, vancomycin-resistant Enterococci (VRE) should be mentioned, especially due to their increasing isolation in hospital settings [21]. In a large study carried out in a non-outbreak setting, the contamination of several environmental sites was found. In particular, 13% of sampled toilet bowl seats were positive for at least an MDR organism (CPE, VRE and/or ESBL-producing Enterobacterales). The rate of environmental contamination was higher for patients with VRE (VanA-producing Enterococci) compared to other MDR organisms. Environmental samples from rooms housing VRE-colonized patients were positive in the 42% of cases [22]. Vancomycin-resistant *Enterococcus faecium* (VREfm) may also shed into the hospital environment, where it may persist despite standard cleaning. Indeed, Noble et al. reported that a hospital toilet represents a transmission vector for VREfm, since it facilitates the patient-to-patient transmission of *E. faecium* [23]. For this reason, being admitted to a room previously occupied by a VREfm-positive patient is a risk factor for acquiring VREfm. Therefore, preventing the *E. faecium* acquisition requires an understanding of reservoirs and transmission routes.

The role of the environment as a source of infection is even more relevant for spore-forming bacteria, such as *Clostridioides difficile*. It has been estimated that a patient with *Clostridioides difficile* infection (CDI) can excrete between 10^4^ and 10^7^ of *C. difficile* per gram of faeces [24]. Indeed, in a study performed by Reigadas et al., the toilet was the second most contaminated site with toxigenic *C. difficile* [25]. The different ribotypes recovered corresponded mostly to frequent ribotypes in Spain [26] (Table 1).

### 3.2. Contamination of Toilet Plumbing Systems

Some studies reported circulation and transmission of CPE originating form hospital plumbing installations [27,28,29,30,31,32,33,34].

A high level of sinks siphons contamination by OXA-48-producing *K. pneumoniae* and *E. coli* was found in a traumatology and oncology ward (Table 1), with patients admitted at rooms contaminated by OXA-48-producing CPE more frequently colonized [29].

A similar scenario involving other CPE was observed in a burn unit in Belgium, where an outbreak of infections by OXA-48-producing *K. pneumoniae* occurred [30]. The concomitant presence of these bacteria in the toilet water of multiple rooms was observed, with the single origin of the outbreak confirmed by whole-genome multi-locus sequence typing (wgMLST) analysis. In fact, all outbreak isolates belonged to the same ST (i.e., ST15) and showed isogenicity (<15 allele differences) [30] (Table 1). Patients were not transferred in other rooms and, conversely, several drainpipe obstructions with consequent water reflux to the different toilets were reported, and thus, it was likely that OXA-48-producing *K. pneumoniae* may have spread between different rooms through the common wastewater plumbing in a retrograde manner. The daily disinfection with bleach for two months was not sufficient to terminate the outbreaks, indicating that this method is insufficient to eradicate the strain in toilet water

Likewise, Hamerlinck et al. found toilet water, sink drains and shower drains mostly contaminated with OXA-48-producing clones (69.9%), followed by VIM (37%) and NDM-1 (1.4%). *C. freundii* was the predominant species (52.1%), followed by *E. cloacae* complex (41.1%), *K. pneumoniae* (9.6%) and *K. oxytoca* (6.8%) [28] (Table 1). Despite infection control measures and appropriate cleaning protocols, a long-term co-existence of five different OXA-48-producing *C. freundii* clusters (ST22, ST170 and ST421, ST481 and ST146) was detected by a retrospective analysis using whole-genome sequencing (WGS) data [28]. Such clusters involved both the patient as well as environmental isolates. Since patient-to-patient transmission was excluded due to long free intervals between stays, sanitary facilities may have played a role in the circulation of these clones.

A recent study in an intensive care unit (ICU) has also demonstrated that sinks traps located near toilets were more frequently contaminated by microorganisms carrying the *bla*_KPC_ gene (i.e., carbapenem-resistant *K. pneumoniae* and *E. cloacae*) than sink drains located away from toilets [31] (Table 1). This phenomenon could be multifactorial and to involve not only biofilm growth in communal pipes between toilets and sinks, but also the generation of contaminated droplets during flushing and/or the conduct of routine hand hygiene by patients or healthcare workers.

Other studies reported sinks as environmental reservoirs of CPE encoding for the *bla*_IMP-4_ (i.e., *E. cloacae* complex ST24, *C. freundii* ST8) [32] (Table 1). In a retrospective outbreak investigation carried out by WGS of stored CPE isolates from Australia, Marmor and colleagues revealed cases of *E. cloacae* complex ST24 among both environmental samples and patients, while cases of *C. freundii* ST8 *bla*_IMP-4_ only among patients [32].

The healthcare wastewater drainage systems (i.e., drains, sink/shower siphons, drainage pipes), can be a reservoir for other pathogens, such as non-fermenting Gram-negative (NFGN) bacteria. Irregular designs of sinks and toilets, frequent blockages and leaks from waste pipes are factors that can contribute to the contamination [35]. This is the case of several outbreaks caused by MDR *P. aeruginosa*, where the hospital wastewater system was identified as a probable source of infection [33,34] (Table 1). Two outbreaks in England, one hospital-wide and the other one limited to a specialized medical unit, were caused by a different *P. aeruginosa* genotype, both producers of VIM-2 carbapenemase [33]. Aspelund and colleagues also described a prolonged nosocomial outbreak of VIM-producing *P. aeruginosa*, where sink drains served as a long-term reservoir. In particular, the majority of strains isolated from sink showed the same antibiotic susceptibility phenotype and were identical or closely related (i.e., they showed the same or very similar pulsed-field gel electrophoresis (PFGE) band pattern) to clinical isolates [34], demonstrating how the hydraulic environment could have played a role in establishing the outbreak.

### 3.3. Contamination of Toilet Flushing-Generated Bioaerosol

In the context of shared toilets, bioaerosols can play an important role in the transmission of infections. When a toilet is flushed, a turbulent flow of water generates aerosols and droplets that can carry microorganisms and spores from faecal matter both into the air and the surrounding environment [12,15,36].

In a pre/post-intervention design study, conducted in adult intensive care units (ICUs) of a university hospital, the installation of hopper covers (a “toilet-like” waste disposal system) and sink trap devices were effective in reducing the acquisitions of carbapenemase-producing Enterobacterales (CPE) through particles dispersion after toilet flushing [37]. The study further confirmed the role of hospital wastewater plumbing as a reservoir in nosocomial transmission of multispecies CPE (e.g., *Serratia marcescens*, *Enterobacter cloacae* complex, *Citrobacter freundii* and *Klebsiella oxytoca*-producing KPC-type carbapenemases) [37].

In another study, reproducing bacterial loads observed during symptomatic phase of *Clostridioides difficile* infection (CDI), Best et al. found airborne spread and consequent environmental contamination of this pathogen through the aerosol produced following flushing in different types of toilets (with and without lid) typically used in hospital settings [38]. The authors detected high levels of *C. difficile*, especially in toilets without lids, after flushing the toilet. The surface contamination near the toilet has also been confirmed [38]. Similarly, the air of bathrooms used by patients with CDI were sampled both before and after flushing by Wilson et al. [39]. The authors found that pre- and post-flush samples were positive for *Enterococcus faecalis*, *E. faecium* and *C. difficile*, with major bacterial concentrations observed in post-flush samples [39].

**Table 1 microorganisms-12-00136-t001:** Features of multidrug-resistant bacteria contaminating shared restrooms.

Reservoir	Hospital Ward ^a^	Pathogen	Molecular Typing ^b^	Resistance Phenotype ^c^	β-Lactamase Genes	Reference
Toilet Bowl, Seat and Rim	Geriatric ward	*Citrobacter freundii*	ST22 (*C. freundii*), ST170 (*C. freundii*), ST421 (*C. freundii*), ST481 (*C. freundii*), ST146 (*C. freundii*)	-	*bla*_OXA-48_, *bla*_VIM_, *bla*_NDM-1_	[19]
		*Enterobacter cloacae* complex				
		*Klebsiella pneumoniae*				
		*Klebsiella oxytoca*				
	Haematology ward	*Citrobacter freundii*	ST22 (*C. freundii*), ST253 (*C. freundii*)	-	*bla* _OXA-48_	[20]
		*Escherichia coli*				
		*Enterobacter sakazakii*				
	Medical ward, surgery, ICU	*Enterococcus faecium*	-	VAN^R^	ESBL, *bla*_OXA-48_, *bla*_KPC_, *bla*_NDM_	[22]
	ICU	*Enterococcus faecium*	-	VAN^R^, TEC^R^	-	[23]
	Gastroenterology, internal medicine, infectious diseases ward	*Clostridioides difficile*	RT001, RT207, RT106, RT050, RT-R164, RT- R186	-	-	[25]
Toilet Plumbing Systems	Psychosomatic, rehabilitation, haemato-oncology ward	*Citrobacter freundii*	ST167 (*E. cloacae*), ST823 (*P. aeruginosa*)	CB^R^, FQ^R^, PIP^R^, CTX^R^, CAZ^R^, CST^R^	*bla*_KPC_ (*K. pneumoniae*), *bla*_VIM_ (*P. aeruginosa*)	[27]
	-	*Citrobacter freundii*	-	-	*bla* _OXA-48_	[28]
	Traumatology and oncology ward	*Klebsiella pneumoniae*	ST11 (*K. pneumoniae*), ST405 (*K. pneumoniae*)	-	*bla*_OXA-48_ (*K. pneumoniae*), *bla*_VIM_	[29]
		*Klebsiella oxytoca*				
		*Enterobacter cloacae*				
		*Serratia marcescens*				
		*Citrobacter freundii*				
		*Raoultella planticola*				
	Burn unit	*Klebsiella pneumoniae*	ST15	AMP^R^, AMC^R^, CXM^R^, CAZ^R^, CTX^R^, TEM^R^, TZP^R^, TOB^R^, LVX^R^, SXT^R^	*bla* _OXA-48_	[30]
	ICU	*Klebsiella pneumoniae*	-	ERT^R^, MEM^R^	*bla* _KPC_	[31]
		*Enterobacter cloacae*				
		*Raoultella planticola*				
		*Raoultella ornitinolytica*				
		*Aeromonas hydrophila*				
	ICU, haematology, renal medicine, surgical ward	*Enterobacter cloacae*	ST24 (*E. cloacae*)	-	*bla* _IMP-4_	[32]
		*Citrobacter freundii*	ST8 (*C. freundii*)			
	Hospital-wide (outbreak 1), haematology ward (outbreak 2)	*Pseudomonas aeruginosa*	-	AG^R^, CB^R^, FQ^R^, ATM^R^, CAZ^R^, TZP^R^	*bla* _VIM-2_	[33]
	Three hospital wards	*Pseudomonas aeruginosa*	-	-	*bla* _VIM_	[34]
Toilet Flushing-Generated Bioaerosol	ICU	*Serratia marcescens* *Enterobacter cloacae* complex *Citrobacter freundii* *Klebsiella oxytoca* *Aeromonas* spp. *Raoultella* sp. Other Enterobacteriaceae	-	-	*bla* _KPC-type_	[37]
	Microbiology department, hospital ward	*Clostridioides difficile*	-	-	-	[38]
	Haematology/oncology, medical speciality, bone marrow transplant, oncology, surgical speciality, orthopaedics/urology ward	*Clostridioides difficile* *Enterococcus faecium* *Enterococcus faecalis*	-	-	-	[39]

^a^ ICU, intensive care unit. ^b^ ST, sequence type (according to the Pasteur MLST scheme); RT, ribotype. ^c^ AMK^R^, resistance to amikacin; AG^R^, resistance to aminoglycosides; AMC^R^, resistance to amoxicillin-clavulanate; AMP^R^, resistance to ampicillin; ATM^R^, resistance to aztreonam; CB^R^, resistance to carbapenems; CTX^R^, resistance to cefotaxime; CAZ^R^, resistance to ceftazidime; CXM^R^, resistance to cefuroxime; CIP^R^, resistance to ciprofloxacin; CST^R^, resistance to colistin; ERT^R^, resistance to ertapenem; FQ^R^, resistance to fluoroquinolones; LVX^R^, resistance to levofloxacin; MEM^R^, resistance to meropenem; PIP^R^, resistance to piperacillin; TZP^R^, resistance to piperacillin-tazobactam; TEC^R^, resistance to teicoplanin; TEM^R^, resistance to temocillin; TOB^R^, resistance to tobramycin; SXT^R^, resistance to trimethoprim-sulfamethoxazole; VAN^R^, resistance to vancomycin.

## 4. Cleaning and Disinfection Methods

Determining effective infection control measures to prevent transmission of MDR microorganisms between patients and the environment is necessary to limit outbreaks. Recent publications have suggested that contact isolation policies and/or individual rooms for colonized or infected patients did not show additional benefits compared to the standard infection control policies [40,41]. Furthermore, conventional protocols often fail to protect patients as well as healthcare workers.

It has been demonstrated that the use of different disinfectants, such as peracetic acid and quaternary ammonium, might reduce environmental contamination [42], although this could increase the risk of resistant bacteria dissemination [43]. Alternatives may be represented by water vapor or gas-plasma, the latter mainly being used to sterilize medical devices rather than sanitary surfaces [44]. Concerning alcohol-based disinfectants, even if effective against most microorganisms, they have scant or no effect against spore-forming bacteria [25]. Chlorine can be effective to remove biofilms from toilet and sink drains, with short exposure times inducing growth inhibition. However, the use of chlorine has been recently associated with the formation of carcinogenic compounds as well as being corrosive and harmful to humans [44]. Finally, ultraviolet (UV) radiation has been increasingly reported as an effective sterilization method, and recently, its efficacy has been verified within a shared hospital bathroom [45].

Overall, intensive cleaning programs, better ventilation inside toilets and an adequate education of healthcare workers together with isolation procedures of infected patients are encouraged.

A more detailed description of chemical products used for toilet sanitation deserves more reviewing activity and is out of the scope of this work.

## 5. Physical Means

Conventional cleaning methods, such as the use of sanitizing chemicals, are often unable to completely eliminate pathogens and protect users.

The adoption of new technologies capable of mechanically preventing the spread of infections after usage of the toilet could be considered as a supplementary measure compared to the traditional methods. A possible technological tool, to be adopted in shared restrooms, is the use of automatic toilet seat covers [46]. Moreover, another containment measure could consist of traditional wall-mounted air vents or air extractors connected to the flush tank. In the first case, air circulation takes a considerable amount of time, and thus, it is not possible to prevent the spread of aerosols, which will contaminate nearby surfaces and be inhaled by the next user. The use of the toilet, in this scenario, should be prohibited until a complete air exchange takes place [47]. In the second case, the extraction is performed through the tube connecting the flush tank to the toilet. However, this system does not effectively extract aerosols because during the flushing process, the air extraction capability disappears completely as the tube becomes filled with water and is unable to extract.

In a feasibility study carried out by Università Cattolica del Sacro Cuore (Rome, Italy), a new medical device, namely “Toilé” (Planus Spa), was proposed. It is a toilet capable of “sucking” aerosol from the toilet bowl during use and carrying it outside the building, in order to remove both odours and potentially pathogen microorganisms [48]. It aims at keeping safe both healthcare workers and patients during the use of toilet. Therefore, such a technology could be used within shared restrooms and environments with a high risk of spreading infections, such as those in healthcare settings. However, there is currently no scientific evidence to support the effectiveness of this medical device. Furthermore, critical issues could be introduced by the complexity of this solution, and its high costs.

An American company [49] sells a powder that, when poured into the toilet vase, forms a foam which is claimed to prevent the formation of droplets, and to cover unpleasant odours and noises. Also, in this case, we are not aware of any published tests that scientifically demonstrate the effectiveness of the product nor of patents that protect the chemical composition.

Finally, our group carried out a study on the physical and microbiological characteristics of some pathogens present in the bathroom, which led to the development of a novel foaming composition. This tool is effective for preventing the generation of droplet rebound during urination by the deposition of a foam layer before the use of the toilet.

In a model simulating female urination, the foam was able to completely suppress the formation of rebound droplets containing a multidrug-resistant strain of *K. pneumoniae*. In particular, this strain belonged to a widely distributed high-risk clone (ST101, sequence-type 101) carrying the *bla*_KPC-3_ and *armA* genes, demonstrating how the use of such a strategy may be important to prevent the spread of clinically relevant infections. It is noteworthy that this foam did not show antibacterial activity in vitro, thus potentially preventing the development of antibiotic resistance [50]. This could be of high relevance for settings where patients with intestinal colonization by carbapenemase-producing *K. pneumoniae* share a room with noncolonized patients (e.g., rehabilitation centres, long-term care facilities). Foams offer several important advantages as they can be deposited onto virtually any surface and can be easily flushed away once urination is completed. Nevertheless, further investigations involving different bacterial species and scalar bacterial loads are needed.

## 6. Conclusions

Healthcare-acquired infections (HAIs) are the most frequent adverse events in hospital settings. A large number of patients contract such infections, leading to higher morbidity and mortality rates and increasing healthcare costs. In this context, the transmission of pathogens among hospitalized patients represents a serious threat, in particular for the presence of multidrug-resistant (MDR) microorganisms contaminating surfaces and environments within patient rooms.

In this review, we revised the literature concerning outbreaks caused by MDR bacteria detected within toilet and plumbing sites therein (i.e., drains and siphons), as a phenomenon underlying HAIs. Overall, the most representative species-resistance mechanism combination is represented by β-lactamases-producing Enterobacterales, regardless of the contaminated toilet site. This is in accordance with widespread evidence that Gram-negative bacteria resistant to β-lactams represent one of the most detected pathogens at the hospital level, especially in an outbreak context. Moreover, the probable correlation between environmental contamination and infection transmission seems to be multifactorial. Firstly, the bioaerosol and droplets generated during the use of the toilet and flushing can represent a hidden infectious source that facilitates the spread of germs in the environment near the toilet. Also, the ability of bacteria to grow in biofilm within sink and/or shower drains might represent a further contamination source. Finally, the routine hand hygiene carried out by patients or healthcare workers together with wrong or inefficient cleaning of hospital environments can lead to the persistence of bacteria in plumbing systems and sanitary installations in bathrooms.

However, this study presents some limitations, including potential publication bias and limited literature sources available. This could be due to a lack of adequate evidence about the role of toilets in the transmission of infections. Furthermore, several studies do not explore possible interventions to prevent spreading of MDR organisms from restrooms and/or to terminate outbreak. Therefore, toilet environments seem not to raise interest in healthcare research and are probably often overlooked in outbreak assessments, which may lead to underestimation of the problem.

Nonetheless, a correlation between bacteria contaminating toilets and bacteria infecting patients was observed in most cases, confirming that the toilet and surrounding areas were the sources of infection. Such a relation was highlighted following phenotyping and genotyping characterization of MDR microorganisms. For this reason, detecting the bacteria contaminating different toilet sites in addition to identifying their molecular features can play an important role in hospital outbreaks, as would making it possible on the one hand to characterize those considered as high-level disseminators, and on the other hand to identify frequent sources of contamination, with the aim being to better target the cleaning of environments and implement appropriate infection control strategies.

## Figures and Tables

**Figure 1 microorganisms-12-00136-f001:**
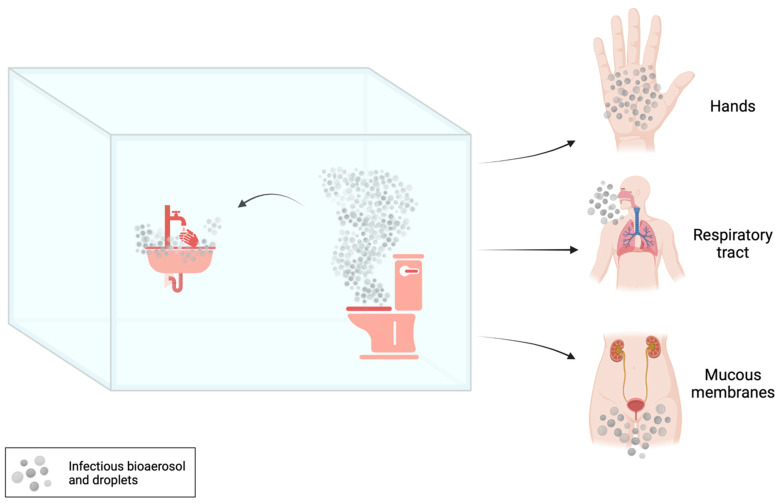
Transmission routes and microbial contamination risks within restrooms. The bioaerosol and/or droplets (grey dots) generated by using toilet (flushing and urination) and sink lead to the dispersion of pathogens potentially inhalable or contaminating environments (i.e., red sections) or human surfaces (i.e., hands, mucous membranes). Created with BioRender.com.

## Data Availability

No new data were created for the manuscript.

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
