# Peer review of "Multidrug-Resistant Bacteria Contaminating Plumbing Components and Sanitary Installations of Hospital Restrooms"

_microorganisms, 2024, doi:10.3390/microorganisms12010136_

Round 1

Reviewer 1 Report

Comments and Suggestions for Authors

The aim of this review is interesting: i) to summarize studies reporting the isolation of multidrug-resistant (MDR) 66 bacteria from hospital restrooms and to describe their molecular traits; ii) to review the 67 existing evidence of transmission between the restrooms environment and patients; iii) to 68 describe effective interventions to prevent MDR dissemination from the restrooms or to 69 terminate outbreaks. Traditional and innovative approaches to reduce contamination 70 risks are also briefly discussed. However, here are my considerations for improving the writing of the manuscript:

- I think that studies regarding MDR bacteria contamination on door handles, soap and alcohol gel dispensers as well as soaps and alcohol gel should be included in the manuscript because they can be a potential source of this type of bacterial contamination.

- Add the limitations of the study in the penultimate paragraph of the manuscript discussion.

Reviewer 2 Report

Comments and Suggestions for Authors

Please see comments in the attached file.

Comments on the Quality of English Language

Very good for non-English speaking authors. Some minor editing required.

Round 2

Reviewer 1 Report

Comments and Suggestions for Authors

I agree with the authors' review of the manuscript and the justification for focusing exclusively on the plumbing systems in the hospital settings.

Reviewer 2 Report

Comments and Suggestions for Authors

This review has improved and is now publishable.

Comments on the Quality of English Language

Still some minor English editing required.